# Changes in *Fusarium* and *Aspergillus* Mycotoxin Content and Fatty Acid Composition after the Application of Ozone in Different Maize Hybrids

**DOI:** 10.3390/foods11182877

**Published:** 2022-09-16

**Authors:** Božana Purar, Ivica Djalovic, Goran Bekavac, Nada Grahovac, Saša Krstović, Dragana Latković, Elizabet Janić Hajnal, Dragan Živančev

**Affiliations:** 1Institute of Field and Vegetable Crops, National Institute of the Republic of Serbia, Maxim Gorki 30, 21000 Novi Sad, Serbia; 2Faculty of Agriculture, University of Novi Sad, Sq. D. Obradovic 8, 21000 Novi Sad, Serbia; 3Institute of Food Technology, University of Novi Sad, Bul. Cara Lazara 1, 21000 Novi Sad, Serbia

**Keywords:** maize, ozone treatment, mycotoxins, fatty acids

## Abstract

Mycotoxins in maize represent a great threat to human health. For this reason, novel technics such as ozone treatment are used to reduce the content of maize mycotoxins. However, there is little knowledge about the effect of ozone treatment on maize quality parameters. This study investigated the changes in *Fusarium* and *Aspergillus* mycotoxins and the changes in fatty acids during the ozone treatment of maize samples. Sixteen maize hybrids were visually tested for the naturally occurring ear rot severity and treated with three different concentrations of ozone (40, 70, and 85 mg/L). Mycotoxin content in maize samples was determined using a high-performance liquid chromatography (HPLC) system, whereas dominant fatty acids were determined using gas chromatography coupled with a flame ionization detector (GC-FID). Ozone treatments could be successfully applied to reduce the content of mycotoxins in maize below the detection limit. Ozone treatments increased the content of monounsaturated fatty acids (MUFAs) and decreased the content of polyunsaturated fatty acids (PUFAs), i.e., linoleic acid (36.7% in relation to the lowest applied ozone concentration), which negatively affected the nutritional value of maize.

## 1. Introduction

Maize is one of the most important crops in the world with an average total production of 887,771 thousand tons in the last decade [1]. In Serbia, maize also appears to be the number one field crop, prevailing over all other crops in both economic importance and sown area. The average of total maize production in the last decade amounted to 6286 thousand tons in Serbia [2]. Maize can be processed into various food, pharmaceutical, textile, cosmetic, and chemical products, as well as feed. However, despite the suitability of the region for maize cultivation, maize grain production is often hampered by the occurrence of toxigenic fungi. Genera *Fusarium*, *Aspergillus*, and *Penicillium* are the most common maize grain fungi found in the field and in maize storage facilities [3]. After infection, maize kernels may be contaminated with mycotoxins, which increase in concentration during preharvest, postharvest, and storage [4].

According to the Food and Agriculture Organization (FAO) estimates, more than 25% of the world’s cereal production is contaminated with mycotoxins [5]. They are quite stable and almost impossible to eliminate from the contaminated material during food processing [6]. Mycotoxin occurrence and concentration mainly depend on growing conditions during the sensitive phases of plant development, hybrid susceptibility, and insect damages. Mycotoxins are the toxic secondary metabolites of fungi, which can be produced under different weather conditions. Drought and extremely high temperatures are important abiotic factors of plant stress, and they directly affect the development of toxigenic fungi and toxin production [7]. Deoxynivalenol (DON) and zearalenone (ZEN) are common in most temperate regions of the world such as Europe and China [8], while aflatoxins (AFs) are prevalent in tropical and subtropical regions because fungus needs temperatures of 32–38 °C for infection [9]. Since the beginning of the 21st century, problems with high mycotoxin content in corn were not registered in Serbia until 2012, which was characterized by extreme drought, and 2014, which was characterized by floods [10]. In addition to weather conditions, plant density and genotype are quite important factors contributing to the occurrence of mycotoxins in maize [11,12]. DON is mostly produced by *F. graminearum* and *F. culmorum* but less frequently by *F. equiseti* and *F. sambucinum* [8], while ZEN is produced by *F. roseum, F. tricinctum*, and *F. sporotrichioides* [13]. Aflatoxins B1 (AFB1) and aflatoxins B2 (AFB2) are mostly produced by *Aspergillus flavus*, whereas aflatoxins G1 (AFG1) and aflatoxins G2 (AFG2) are mainly produced by *A. parasiticus* [14].

Mycotoxins have different effects on human and animal health. DON greatly affects the intestine and the immune system of pigs [15], decreasing feed consumption at doses ranging from 0.6 to 3 mg DON/kg feed [16]. In humans, DON causes diarrhea, abdominal pain, headache, dizziness, and fever [17]. Although ZEN has lower acute toxicity in comparison with many other mycotoxins (oral LD 50 values of >2000–20,000 mg/kg b.w. in mice, rats, and guinea pigs), it is a powerful estrogen whose hormonal activity exceeds that of most other naturally occurring nonsteroidal estrogens [18]. Aflatoxin B1 is more dangerous because it can provoke cancer in rat liver [19] with a reported LD_50_ for the adult male Fisher rat of 1.2 mg aflatoxin B1/kg body weight [20]. Other mycotoxins in the aflatoxin group are also carcinogenic [21]. Numerous studies conducted worldwide confirmed the presence of the abovementioned mycotoxins in maize [13,22,23,24,25,26,27,28]. Considering the occurrence of mycotoxins, as mentioned above, it is not possible to obtain safe products with standard processing of the contaminated maize. Therefore, new alternative techniques for maize decontamination, such as thermal processing [29], chemical agents, food additives, and biological detoxification and biotransformation, are currently being investigated [30]. Among the investigated techniques, it seems that the ozone treatment could be an efficient method of maize grain decontamination [31].

Ozone is a strong oxidant that can react with many contaminants in water. Due to this property, ozone is suitable for pollutant removal systems for treatment of drinking water and wastewater [32]. It is typically generated in an ozone generator, a silent corona discharge unit that uses electrical discharge on oxygen molecules to release free radicals of oxygen and ultimately cause the formation of ozone [33]. In addition to application in water treatments, ozone has been used for decontamination of different food materials, such as cereal grains [34,35,36,37,38]. McKenzie et al. [39] found that ozone effectively degraded mycotoxins (AFs, fumonisins, ochratoxin A (OTA), and ZEN) under in vitro conditions, with degradation products which were generally nontoxic. Wang et al. [38] managed to reduce DON content in wheat using ozone at level of 75 ppm, obtaining a 53.48% reduction rate after 90 min of exposure. Santos Alexandre et al. [35] reported DON and ZEN reductions (32% and 61%, respectively) in wheat bran after treatment with 62 mg/L ozone for 240 min. Sun et al. [36] applied aqueous ozone for 10 min at an ozone level of 80 mg/L, managing to reduce DON content in wheat, corn, and bran by up to 74.86%, 70.65%, and 76.21%, respectively. The study of Qi et al. [40] indicated that ozone treatment was effective for ZEN and OTA decontamination of maize, after applying ozone at the level of 40–100 mg/L for up to 180 min. Furthermore, some studies have found that ozonization can cause changes in grain quality attributes, such as protein [38], fatty acids [35,40], rheological properties [41], color [40], phenolic compounds, and antioxidant ability [35], especially in cereals and its products.

The aim of the study was to investigate the natural mycotoxin contamination of maize grain in a set of commercial maize hybrids and the possibility of using ozone for grain decontamination. Furthermore, the aim of the study was to evaluate the effect of ozone treatment on grain dominant fatty acids.

## 2. Materials and Methods

### 2.1. Field Trial

The field trial was set up in the experimental fields of the Institute of Field and Vegetable Crops, Novi Sad, Serbia in 2021 (45°20′14″ N, 19°51′44″ E, 78 m above sea level). The climate of the region is moderately continental, and autumn is drier than spring. The warmest month is July and rainiest month is June. Annual precipitation varies between 570 and 650 mm. The experiment was carried out on a chernozem soil, with a humus content of 2.86%, pH = 7.04, total nitrogen 0.24%, P_2_O_5_ 27.53 mg/100 g, and K_2_O 29.23 mg/100 g. The preceding crop was soybean in a 3 year rotation (soybean–maize–wheat), and conventional soil cultivation practices were applied. Sowing was conducted on 14 April, pre-emergence herbicide was applied 5 days later, and post-emergence correction was applied at the 6–7 leaf growth stage. A total of 16 maize hybrids of different genetic background, designated H1–H16, were used as entries in a randomized complete block design (RCBD) with three repetitions. Four row plots (10 m long and 0.75 m wide) were planted at an approximate density of 63,500 plants/ha.

### 2.2. Harvest, Assessment, and Sampling

At physiological maturity, ears from each plot were de-husked and harvested manually. The naturally occurring ear rot severity was assessed visually according to Reid and Zhu [42]. However, the original scale was modified to a four-point scale: 1 = 0–10%, 2 = 11–35%, 3 = 36–75%, and 4 = 76–100% of kernels exhibiting visual symptoms of infection (presence of white, pink, grey, green or reddish mycelia). Then, 40 ears of each plot were air-dried and shelled, the grains were thoroughly mixed, and samples of 2 kg were taken for mycotoxin analysis.

### 2.3. Sample Preparation

Before mycotoxin and free fatty acid analysis, grains of hybrid samples were left in ambient conditions until the grain moisture dropped below 12%. Moisture content was determined using the American Association of Cereal Chemists (AACC) method 44–15.02 [43]. Using the quartering procedure, the samples were reduced to subsamples [44] approximately 1 kg in weight, and then ground in the laboratory mill (IKA A11 basic, Staufen, Germany).

### 2.4. Chemicals

All solvents used in this study were of analytical grade and obtained from Fisher Scientific (Fair Lawn, NJ, USA) and Sigma-Aldrich (St. Louis, MO, USA). HPLC-grade acetonitrile, methanol, trifluoroacetic acid, and *n*-hexane were purchased from Sigma-Aldrich (St. Louis, MO, USA). Fatty acid methyl ester standards (RM-1, FAME), as well as analytical standards of mycotoxins, were obtained from Supelco (Bellefonte, PA, USA). The methanolic trimethylsulfonium hydroxide (TMSH) solution was purchased from Merck (Darmstadt, Germany).

### 2.5. Ozone Generation and Maize Decontamination Procedure

Gaseous ozone was generated by the corona discharge ozone generator MOG002 (O3 Tech H.K Limited, Shenzhen, China) using medical oxygen. The ozone concentration was regulated by varying oxygen flowrate, and ozone levels of 40, 70, and 85 mg/L were used in this research. Similar ozone levels were applied by Wang et al. [37] and Qi et al. [40]. Ozone application was performed by placing approximately 30 g of finely ground maize into a glass cylinder (150 mm long and 20 mm in diameter) with two apertures (bottom for ozone input and top for output). Ozone gas was run for 180 min, and ozonized maize samples were immediately collected and identified. Mycotoxins were extracted within 1 h of ozone exposure, and the extracts were kept in the refrigerator until analysis. All decontamination experiments were conducted at room temperature in duplicate.

### 2.6. Mycotoxin Determination

DON, ZEN, and AF determinations were carried out on a 1260 series HPLC system (Agilent Technologies, Santa Clara, CA, USA) with DAD and FLD detectors (Agilent Technologies, Santa Clara, USA) and a Hypersil ODS (150 × 4.6 mm I.D., particle size 5 μm) column (Agilent Technologies, Santa Clara, CA, USA). Specifically, 12.5 g samples were extracted using 50 mL of an acetonitrile and water mixture (84:16, *v*/*v*). The extracts were then cleaned up on Mycosep^TM^ 224, Mycosep^TM^ 225, and Mycosep^TM^ 229 columns (Romer Labs. Inc., Union, MO, USA), and 3 mL of cleaned up extract was evaporated until dryness at 60 °C under a gentle stream of nitrogen. The residue was dissolved in 300 µL of mobile phase. For AF determination, a derivatization step using trifluoroacetic acid was also carried out [45]. HPLC conditions for DON were set as proposed by Abramović et al. [46]. For AFs, the method by Oliveira et al. [45] was used, while ZEN was determined according to British standard (BS) EN 15792:2009 [47]. All analyses were conducted in duplicate. Method validation and analytical quality control were described elsewhere [31]. Limits of detection (LODs) were 0.3 µg/kg for AFs, 1.1 µg/kg for ZEN, and 22.2 µg/kg for DON. Limits of quantification (LOQs) for AFs, ZEN, and DON were 1.0 µg/kg, 3.6 µg/kg, and 74.0 µg/kg, respectively. To test the accuracy of the DON and ZEN determination methods, a maize certified reference material labeled BRM 3024 (Romer Labs, Tulln an der Donau, Austria) was used. Another maize certified reference material labeled TR-A100 (Trilogy, Washington, MO, USA) was used for AFs. The average value for the trueness of the method was 92.3% for DON and 95.7% for ZEN, while the average trueness for AFs was 105.6%.

### 2.7. Seed Oil Determination by Soxhlet Extraction Method

Seed oil content was determined according to the Association of Official Agricultural Chemists (AOAC), method No. 920.85 [48], using a Soxhlet apparatus (Soxtherm 2000 automatic, Gerhardt, Königswinter, Germany), according to the manufacturer’s instructions. The maize seeds were ground using an IKA mill (A11 basic, Staufen, Germany) and sieved into 1 mm fractions. The seed powder was packed in a tumble, and the oil was extracted for 8 h using petroleum ether (boiling point 40–65 °C). The oil content of samples was expressed in g/100 g of maize seeds (dry weight).

### 2.8. Fatty Acid Composition Analysis

Fatty acid (FA) composition was determined using a gas chromatograph (Konik HRGC 4000) coupled with a flame ionization detector (GC-FID) after derivatization to volatile FA methyl esters (FAMEs). FAMEs were prepared by chemically converting FAs to FAMEs according to American Oil Chemists’ Society (AOCS) official method No. Ce 2-66 [49], with some modifications. Briefly, sample oil (10 µL) was exposed to transesterification using TMSH solution (190 µL, 0.2 mol/dm) and directly added to the oil. The reaction vial was capped and strongly shaken in the vortex (IKA, Staufen, Germany) for 1 min. The reaction was complete upon dissolution of the oil. After 1 h, a 1 µL (split ratio 1:70) aliquot of prepared FAMEs was taken for GC-FID analysis. A fused silica capillary column Omegawax 250 (30 m length, 0.25 mm ID, and 0.25 µm film thickness) was used. The column temperature was programmed from 150 °C to 250 °C at 12 °C/min with 8 min holding time. The detector and injector temperatures were set at 250 °C. The carrier gas was helium with a constant flow rate of 1 mL/min. Identification and quantification of individual FAMEs was performed by comparing the GC retention times and the Kovats retention index [50] with the pure commercial standard mixture of FAMEs (Supelco FAME, palmitic acid (C16:0), stearic acid (C18:0), oleic acid (C18:1), linoleic (C18:2), α-linolenic acid (C18:3), and arachidic acid (C20:0)) and analyzed under the same conditions. Identifications were based on a comparison of the GC retention times (Kovats’ index) with reference analytical standards. The results of the individual fatty acid content were expressed as g/100 g of oil.

### 2.9. Statistical Analysis

#### 2.9.1. Ear Root Assessment

Ear rot assessment scale consists of four categories with natural ordering (i.e., category ranking 1 < category rating 1 < … < category rating 4). When the response variable (*Y*) has natural ordering, the parametric statistical model can be used to exploit various logit transformations of the response probabilities [51,52]. If we assume that the response categories from ear rot scale have spacing on a continuous scale, category rating 1 will represent the plants as *Y* < *θ*_1_, where *θ*_1_ represent the first cut-point. The probability (*π*_1_) of observing the plants with category 1 is *P*(*Y* < *θ*_1_). Category rating 2 represent the plants between *θ*_1_ and *θ*_2_, and the probability (*π*_2_) of the observing the plants with this response category is defined as *P* (*θ*_1_ < *Y* < *θ*_2_). As the ear rot assessment scale has four category ratings, then the model will estimate three cut-points (*θ*_1_, …, *θ*_3_), three cumulative probabilities (*γ*_1_, …, *γ*_3_), and four estimated probabilities (*π*_1_, …, *π*_4_).

Modeling of the cumulative probabilities was based on the following ordinal cumulative logit model: *θ*_i_ = 1/[1 + e^−(*θ*i − β1*X*-…)^], where *θ*_i_ are cut-points, and *X* is the hybrid factor. Prior to estimation of the model parameters, we assumed that the logit of the cumulative probabilities changed linearly as the predictor variable changes, and that the slope was the same regardless of the category *i*. The unknown parameters of the cumulative logit model were estimated using the maximum likelihood method. The statistical inference of the estimated parameters was performed using Wald’s test, which, under the null hypothesis, has an approximate chi-square distribution [53].

#### 2.9.2. Fatty Acid Data

The fatty acid compositional data of the studied maize hybrids were analyzed using the compositional data analysis methodology [54]. This methodology is related to the positive and non-Gaussian data carrying relative information. Values in compositional data are dependent because, as one value increases or decreases, the others increase or decrease collectively by the same amount [54]. Moreover, when data are compositional, estimation of the correlation is arbitrary, and classical mean and variance cannot be used to properly describe the central tendency and spread of the data [54,55]. In our study, the association among the fatty acid compositional parts was estimated through a variation array, which is the compositional data analysis counterpart of correlation analysis [54]. For example, in a variation array, the variation of the C16:0 and C18:0 is defined as the variance of the log (C16:0/C18:0) ratio. When this ratio value is low, the association is high and vice versa.

Principal component analysis (PCA) is a widely used multivariate technique for data reduction analysis. Unlike correlation analysis, the PCA possesses pitfalls of the classical analysis conducted on compositional data, including lower explained variance and a distorted relationship among the compositional parts and the samples. Several data transformations can be readily applied on compositional data. Among them the centered log ratio (clr) transformation is suitable for the purpose of the PCA. The clr transformation is the natural log of the observation divided by the geometric mean of all compositional parts for that observation [54]. Application of the PCA technique is frequently violated by the presence of the outlier observations. For the purpose of this study, a robust variant of the clr PCA for compositional data was applied [56].

## 3. Results and Discussion

### 3.1. Ear Rot Assessment

Maize grain contamination with mycotoxins depends on the environmental conditions during cultivation, hybrid resistance to contamination, and the interaction between both factors [57]. Very warm and rainy July (heat wave from 6 to 16 July and precipitation above 90 mm), and warm and dry August (heat wave from 25 July to 2 August and precipitation 35 mm) in 2021 [58] are favorable for ear rot development. Although all genotypes are affected under favorable conditions, hybrids differ significantly in terms of ear rot resistance and mycotoxin content [59].

Our study contained three standard hybrids (H4, H8, and H16) and three independent models with these hybrids as the reference levels, enabling direct comparison between the reference and the remaining studied hybrids (Figure 1). Hybrids H13 and H14 were highly and significantly different (*p* < 0.01) from the reference hybrids in terms of ear rot. On the other hand, hybrids H1 and H12 were not different (*p* > 0.05) from the reference hybrids in view of ear rot.

Probabilities and cumulative probabilities of the hybrid classification are depicted in Figure 2. The first cumulative probability for H1 was 0.441. This indicates the probability of hybrid H1 receiving a category rating not larger than 1. Similarly, the second cumulative probability for this hybrid was 0.769, which can be interpreted as probability of receiving a category rating not larger than 2. Estimated probability denotes the average probability of assignment of a hybrid into a given category rating. Except for H2, all studied hybrids could be classified (assigned) into category rating 1 and, thus, considered as resistant to ear rot (Figure 2).

### 3.2. Effectiveness of Ozonation on Mycotoxin Levels

Mycotoxin analysis in maize hybrids revealed a high incidence of DON (15/16 samples above the LOQ) at low to medium levels. ZEN was not quantified in any of the tested maize hybrids, while aflatoxins were present in seven samples (Table 1).

As expected, AFB1 was the most prevalent mycotoxin among aflatoxins at levels from 1.1 ± 0.0 to 328.5 ± 2.5 µg/kg. Moreover, five of the seven positive samples of maize grown for human consumption exceeded the maximum level (ML) for AFB1 and the sum of aflatoxins regulated by the EU [60]. On the other hand, only one of 15 positive samples contained DON above the ML prescribed by the EU regulation [60]. Large differences in terms of mycotoxin content could be attributed to genetic differences among hybrids (G) and specific interaction between hybrids and the environment (GxE), since all hybrids were produced in the same conditions (planting conditions, temperature, moisture, etc.). To assess the effect of ozone, samples containing high levels of mycotoxins were selected, as well as samples contaminated with single or multiple mycotoxins. Therefore, we chose H1 containing DON and AFs at high levels (above regulated ML), as well as H2 and H12 containing only DON near the ML. H5 and H11 were selected to assess low DON, as well as high AFB1 levels. Lastly, hybrid H14 was chosen so as to evaluate the ozone effect on the samples containing aflatoxins only. Ozonation at three ozone levels (40, 70, and 85 mg/L) applied for 180 min resulted in mycotoxins decreasing below the LOD in all tested samples. Maize samples could, thus, be considered mycotoxin-free. To the best of our knowledge, this is the first report on the complete reduction in AFs and DON in ground maize samples by ozonation. Moreover, the chromatograms of maize sample before and after ozonation revealed no residual peaks (Figure 3).

So far, only limited information is available on mycotoxin (DON, ZEN, AFs, and OTA) reduction in maize using ozone treatment [61]. In the study conducted by Sun et al. [35], DON content in maize was reduced up to 70.7% via application of aqueous ozone for 10 min at ozone level of 80 mg/L. In contrast, via application of dry ozone at the level of 85 mg/L for 180 min, Krstović et al. [31] reported reduction in DON up to 42.8% in a ground maize sample. It is important to note that the initial concentration of DON in the ground maize sample in the study by Krstović et al. [31] was quite high (11.3 mg/kg), 10 times higher than that in our study (1.4 mg/kg). This fact may explain the complete degradation of DON in our study, implying that in addition to all factors that affect the success of ozone treatment (ozone concentration, exposure time, state of ozone, structure of mycotoxins, moisture content in the matrix, matrix weight, and specific surface area of the matrix) for DON degradation in maize, an important factor is also the initial concentration of DON in the sample. The results of this study indicate that complete degradation of DON in ground maize is possible if the initial concentration of DON in the sample before ozone treatment is close to or below the ML [60]. Regarding the reduction in AFs of maize flour samples by means of ozonation, in the study performed by Luo et al. [62], after treatment of maize flour with 75 mg/L ozone for 60 min, the initial concentrations of AFB1, AFG1, and AFB2 of 53.6, 12.1, and 2.4 µg/kg were reduced by 78.8%, 72.1%, and 70.7%, respectively. Furthermore, in recent research on the detoxification of AFG1, AFB1, AFG2, and AFB2 in 1 kg of artificially contaminated maize grit sample (concentration of 50 µg/kg of each AF in sample), the greatest reductions in their content of 54.6%, 57.0%, 36.1%, and 30.0%, respectively, were obtained by ozonation with 60 mg/L of ozone for 480 min [63]. In contrast to the above findings, in the present study, ozonation with 40, 70, and 85 mg/L ozone with an exposure time of 180 min, regardless of the initial concentration of AFs in ground maize samples, resulted in the content of AFs decreasing below the LOQ (DON (74.0 μg/kg), ZEN (3.6 μg/kg), AFs (1.0 μg/kg)) of the applied analytical method in all tested samples (H1, H5, H11, and H14).

### 3.3. Fatty Acids

The quality of grain oil is defined by shares of unsaturated and saturated fatty acids (SFAs). According to Sanjeev et al. [64], in maize, unsaturated fatty acids (C18:1 and C18:2) are dominant, whereas the amount of SFA is much lower. The dispersion of fatty acid composition was shown in a variation matrix (Figure 4a). The C16:0 and C20:0 showed the largest variations and weakest associations with unsaturated fatty acids. Furthermore, association among SFAs was the strongest since covariation between their pairs was low. To gain insight into the effect of ozone treatment on changes in fatty acid compositions, the results were subjected to robust compositional PCA. The first principal component (PC1) explained more than 80% of the variability, whereas the second principal component (PC2) explained about 13% of the variability (Figure 4b). PC1 showed a positive correlation with SFA and a negative correlation with unsaturated fatty acids. Moreover, according to PC1 in the factorial plane, control samples of maize hybrids were clearly separated from samples of ozone-treated samples. Less pronounced differentiation was observed among applied ozone concentrations. Most samples treated with 40 and 70 mg/mL ozone were positioned on the right side of the factorial plane, whereas half of the samples treated with 85 mg/mL ozone were positioned on the left (H2, H11, and H12) and right sides (H1, H5, and H14). Additionally, control samples of maize hybrids were gathered around high concentrations of unsaturated fatty acids. This is important since the nutritional value of oil depends on the content of unsaturated fatty acids, especially polyunsaturated fatty acids (PUFAs), which play an important role in regulating blood cholesterol and lowering blood pressure [65].

The relative amounts of unsaturated and SFAs in the milled maize samples affected oil quality. Figure 5 represents the total content of saturated fatty acids in maize samples treated with different ozone concentrations (ozone 40, 70, and 85 mg/mL) and the control samples (ozone 0). An increasing trend of SFA was observed in all tested hybrids for the lowest applied ozone concentration of 40 mg/L (from 3.0 to 6.8 g SFA/g oil). In the case of ozone concentration of 85 mg/L, it was determined that the content of SFA decreased by 52%, in relation to the share determined by applying the lowest ozone concentration. A decline in monounsaturated fatty acid (MUFA) content was observed for all tested hybrids with the applied treatments, except for hybrid H1.

The PUFA content (C18:2) in the analyzed maize hybrids (H1, H11, and H14) treated with the lowest applied concentration of ozone decreased by more than 5.8 g SFA/g oil compared to the control samples, while, for the other applied concentrations of ozone (70 and 85 mg/L), an increase in the C18:2 content (36.7% in relation to the lowest applied ozone concentration) was observed in the analyzed maize hybrids (H1, H11, and H14). A slight decrease in PUFA was observed for the other analyzed maize hybrids. The amount of linolenic acid did not change significantly during the ozonation process in the ozonized maize hybrids (ozone 40, 70, and 85), compared to the control samples (ozone 0). As mentioned above, a high content of PUFA (2–3 g/day) has positive effects on human health; therefore, the increase in MUFA and decrease in PUFA during ozonation had a negative effect on the nutritive value of maize samples [66].

The content of unsaturated fatty acids (FAs) (MUFAs and PUFAs with two double bonds) was extremely variable when reacting with ozone. Only the total SFAs, such as C16:0 and C18:0, increased during the ozonation process. However, all other examined FAs (C18:1, C18:2, and C18:3) were mostly decreased after the ozone treatment, especially C18:2, when the lowest concentration of ozone (40 mg/L) was applied. Obadi et al. [67] also reported slightly lower concentrations of C18:2 and linolenic acids and an increase in C16:0 after the ozone treatment in the oil extracted from whole-grain flour. This increase in C16:0 could be because of PUFA degradation induced by the ozone treatment. Furthermore, the results were in accordance with the study by Alexandre et al. [41], who observed an increase in SFA and a decrease in unsaturated FA in maize flours during the ozone treatment. There may have been a cleavage of double bonds or PUFA ozonolysis (e.g., C18:2 and C18:3), which led to an increase in SFA content at different ozone concentrations. Numerous factors mentioned by Choe and Min [68], such as concentration and type of oxygen and free fatty acids, mono- and diacylglycerols, pigments, antioxidants, oil processing, energy of heat or light, transition metals, peroxides, and thermally oxidized compounds, jointly affect oil oxidation. Furthermore, this effect can be attributed to the presence of enzymes such as highly active lipoxygenase, which uses ozone to catabolize PUFA [69].

## 4. Conclusions

To the best of our knowledge, this is the first study conducted in Serbia that examined the changes in the quality parameters of maize such as fatty acid content after ozone treatment. Sensory ear rot evaluation showed that almost every maize hybrid, except for H2, could be considered as resistant to ear rot. Results of ozone application indicated that, for all the determined mycotoxins in the maize samples, the ozone treatment (40, 70, and 85 mg/L) successfully reduced mycotoxin content to values below the LOQ (DON (74.0 µg/kg), ZEN (3.6 µg/kg), and AFs (1.0 µg/kg)). On the other hand, ozone application showed a negative effect on the dominant fatty acids of maize samples. An increase in MUFA and a decrease in PUFA (C18:2) were noted, especially when the lowest concentration of ozone (40 mg/L) was applied. However, these differences could be dependent on the genotype; thus, the research in this area should continue. Moreover, further research needs to be focused on the optimization of ozonation (ozone level and reaction time) to minimize the negative impact on the dominant fatty acids in maize while sustaining mycotoxin decontamination efficacy.

## Figures and Tables

**Figure 1 foods-11-02877-f001:**
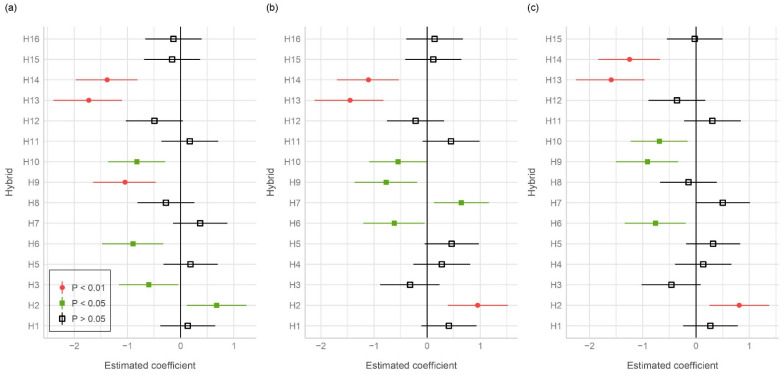
Comparison among the standard reference hybrids with remaining studied hybrids: (**a**) comparison with hybrid H4; (**b**) comparison with hybrid H8; (**c**) comparison with hybrid H16.

**Figure 2 foods-11-02877-f002:**
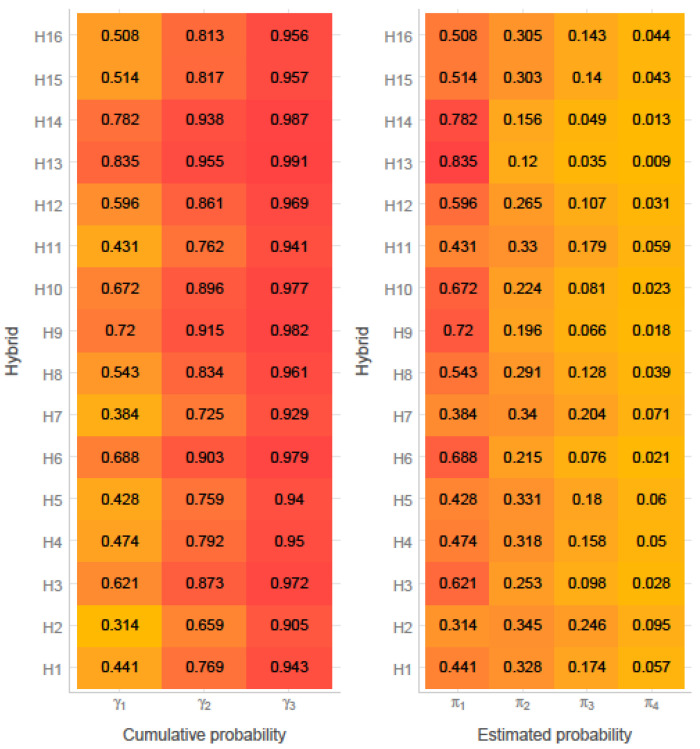
Cumulative and estimated probabilities from the ordinal model based on ear rot assessment scale.

**Figure 3 foods-11-02877-f003:**
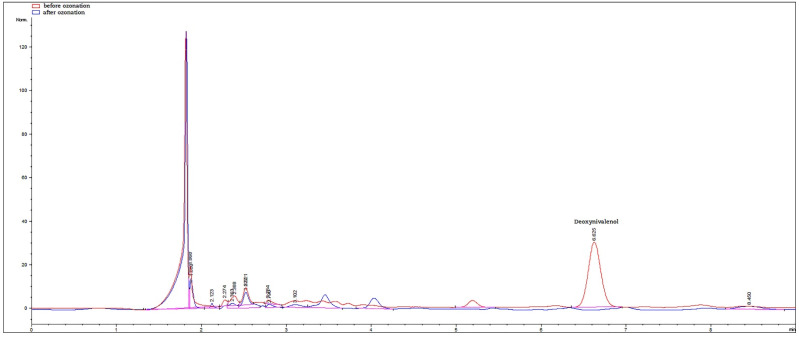
Overlaid chromatogram of DON in maize sample (H2) before (red line) and after (blue line) ozone treatment (180 min at ozone level of 85 mg/L).

**Figure 4 foods-11-02877-f004:**
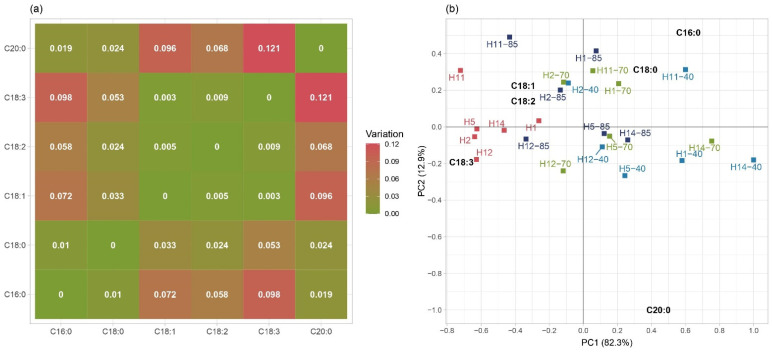
Variation matrix associations among the maize fatty acids (**a**) and compositional robust principal component analysis biplot (**b**).

**Figure 5 foods-11-02877-f005:**
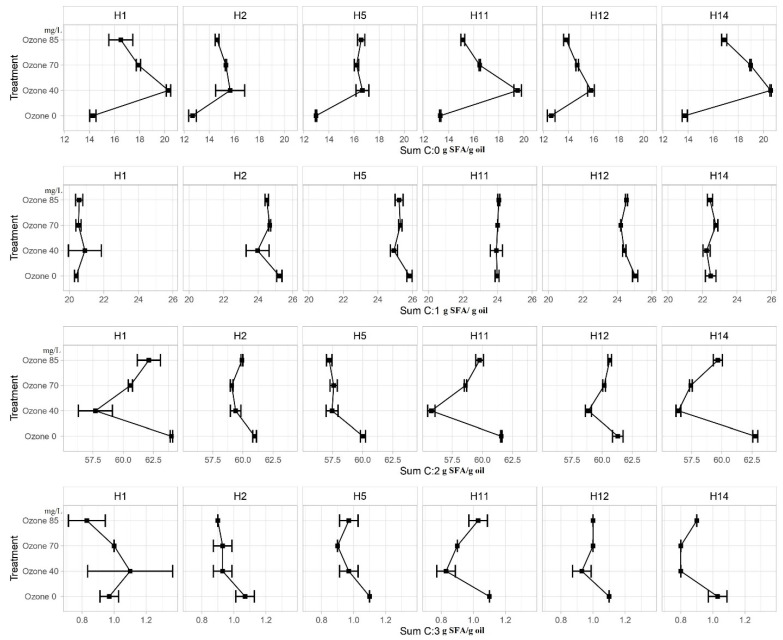
SFA total content of ozone-treated maize samples with different ozone concentrations compared to the control samples (g SFA/g oil).

**Table 1 foods-11-02877-t001:** Mycotoxin content (average ± standard deviation) in maize hybrids before ozonation (*n* = 4).

Maize Hybrid	DON (µg/kg)	ZEN (µg/kg)	AFB1 (µg/kg)	AFG1 (µg/kg)	AFB2 (µg/kg)	AFG2 (µg/kg)	AFB1 + AFG1 + AFB2 + AFG2 (µg/kg)
H1	1399.5 ± 3.5	<LOQ	98.6 ± 2.7	5.4 ± 0.1	9.2 ± 0.1	<LOQ	113.2 ± 2.8
H2	839.0 ± 2.0	<LOQ	<LOQ	<LOQ	<LOQ	<LOQ	<LOQ
H3	656.0 ± 12.0	<LOQ	<LOQ	<LOQ	<LOQ	<LOQ	<LOQ
H4	403.0 ± 17.0	<LOQ	<LOQ	<LOQ	<LOQ	<LOQ	<LOQ
H5	321.0 ± 4.0	<LOQ	89.2 ± 6.9	3.4 ± 0.2	4.0 ± 0.2	<LOQ	96.5 ± 6.9
H6	353.0 ± 16.0	<LOQ	<LOQ	<LOQ	<LOQ	<LOQ	<LOQ
H7	362.0 ± 1.0	<LOQ	2.3 ± 0.1	<LOQ	<LOQ	<LOQ	2.3 ± 0.1
H8	340.0 ± 14.0	<LOQ	<LOQ	<LOQ	<LOQ	<LOQ	<LOQ
H9	444.0 ± 11.0	<LOQ	<LOQ	<LOQ	<LOQ	<LOQ	<LOQ
H10	350.0 ± 9.0	<LOQ	1.1 ± 0	<LOQ	<LOQ	<LOQ	1.1 ± 0
H11	467.5 ± 9.5	<LOQ	328.5 ± 2.5	5.4 ± 0.1	23.7 ± 0.4	<LOQ	357.6 ± 2.2
H12	746.0 ± 18.0	<LOQ	<LOQ	<LOQ	<LOQ	<LOQ	<LOQ
H13	533.0 ± 22.0	<LOQ	<LOQ	<LOQ	<LOQ	<LOQ	<LOQ
H14	<LOQ	<LOQ	43.4 ± 1.8	<LOQ	2.7 ± 0.1	<LOQ	46.1 ± 1.9
H15	487.0 ± 6.0	<LOQ	8.9 ± 0.8	<LOQ	1.2 ± 0.1	<LOQ	10.0 ± 0.7
H16	400.5 ± 6.5	<LOQ	<LOQ	<LOQ	<LOQ	<LOQ	<LOQ

LOQs: DON (74.0 µg/kg), ZEN (3.6 µg/kg), AFs (1.0 µg/kg).

## Data Availability

The data presented in this study are available on request from the corresponding author.

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
