# Peer review of "Changes in Fusarium and Aspergillus Mycotoxin Content and Fatty Acid Composition after the Application of Ozone in Different Maize Hybrids"

_foods, 2022, doi:10.3390/foods11182877_

Round 1

Reviewer 1 Report

Reviewer comments

In this research article the authors focused on the mycotoxin content and individual fatty acids content of 16 maize hybrid samples, after ozone treatment. A current pertinent subject that aims to reveal the potential of ozone treatment in the control of mycotoxin production in maize samples. The subject was well chosen, the manuscript has a careful presentation and its moderately well-written. To improve the quality of the manuscript, some modifications (format, English language, missing information) must be taken in account. Therefore, in the following table some modifications are proposed to polish the final version.

Text Location

Comment

Lines 50 and 51 in Introduction section

Consider indicating the range of temperatures after temperate regions, warm regions and high soil temperatures.

Line 55 and line 57, in introduction section

Indicate the doses of toxicity for the different mycotoxins and add the corresponding reference

Line 70, in introduction section

Since it was previously applied to cereal grains, some success examples can be added. Give indication of the type of cereal grain and in which concentration the ozone has been applied.

Line 73, in introduction section

Which quality parameters were minimally affected. Could you be more specific regarding the term “minimally affected”

Line 75, in introduction section

A paragraph regarding previous studies on the impact of ozone in cereals quality parameters, in general, and fatty acids should be considered to introduce the relevance of studying the fatty acid profile in the present study.

Line 83, in materials and methods

Add a reference for the “common agriculture practice”

Line 89, in materials and methods

Could you give reasons for changing the original scale? (Such as statistical analysis). Is it possible to support the four-point scale with a figure to allow future repetition of the kernels' visual appreciation?

The 0 should be considered separately. Can you support the inclusion of 0 in the first point of the scale?

Line 108, in materials and methods

Consider changing flowrate to flow rate

Line 109, in materials and methods

Justify the applied ozone concentrations and/or add references

Line 121, in materials and methods

Consider changing “evaporated just to dryness” to evaporated until dryness

Line 125, in materials and methods

Consider adding information regarding the compounds’ recovery percentage using the extraction procedure and indicate the LOD and LOQ for the different evaluated Afs. Add the reference, if necessary

Line 140, in materials and methods

Consider removing “was” before directly added to the oil. Substitute “by the vortex” to in the vortex.

Line 141, in materials and methods

Re-write “after for an hour standing” to after one hour standing, the aliquot….

Line 147, in materials and methods

Reinforce the identification by adding the Kovat index of the compounds

Re-write the sentence “The results were expressed…” to The results of the individual fatty acids content were expressed as g/100g of oil Please confirm if I understood correctly and change accordingly

Line 177, in materials and methods

Punctuation is missing after the reference [45]

Line 180, in materials and methods

The principal component analysis (PCA) is a widely used multivariate technique for data reduction analysis. Substitute problems by analysis.

Line 185, in materials and methods

Add reference to the clr transformation explanation

Lines 191 and 192, in results and discussion

Add the range of temperatures and precipitation levels in July and August 2021

Line 192, in results and discussion

Add a comma before “all genotypes are affected…”

Line 194, in results and discussion

Put “in our study” between commas

Line 195, in results and discussion

Consider to substitute “This enables us” by This enabled the direct comparison….

Line 196, in results and discussion

Substitute “rest of studied hybrids” by remaining studied hybrids (Figure1).

Caption of Figure 1

Substitute “rest of studied hybrids” by remaining studied hybrids

Line 207, in results and discussion

Add a comma after Except for H2, ….

Line 213, in results and discussion

Give reasons for not quantifying ZEN.

Table 1, in results and discussion

To improve the table’s clarity, indicate the meaning of the different abbreviators applied in the table caption. 

In the table caption indicate after maize hybrids (H1 - H16) and give an indication regarding the LOQ value (if necessary, add a reference).

Add the units in the last column

Revise the decimals throughout the table.

Indicate in the table caption what we are looking at (average ± standard deviation???) and the number of replicates (n=???)

Line 220, in results and discussion

Consider improving the text by adding some references of previous studies reporting the levels of mycotoxins in maize. Indicate the results’ agreement or disagreement to previous studies

Line 225, in results and discussion

Put below the LOQ between commas and indicate the LOQ value.

Line 226, in results and discussion

Consider reformulating the sentence “maize samples can be considered mycotoxins-free” based on values lower than LOQ. If the values are below than LOQ but higher than LOD the affirmation is incorrect.

Add “the” in the sentence: “To best of our knowledge this first report on the complete…”

Line 235, in results and discussion

Consider changing the percent values with two decimals for only one decimal and change throughout the text accordingly. Add a comma after the percent 70.7%.

Line 238, in results and discussion

Confirm if you want to stay grand or ground. Put in the study by …[26] between commas

Line 239, in results and discussion

Be consistent in the number of decimals used to report the data (two, three, or four decimals). Correct and reformulate accordingly throughout the text.

Line 244, in results and discussion

If possible, add a reference

Line 247, in results and discussion

Correct the number of decimals in the percentages

Line 249, in results and discussion

Remove “of” before 54.6% and add a comma. Also add a comma after respectively

Line 252, in results and discussion

Indicate the LOQ value

Line 258 and 259, in results and discussion

For readers elucidation please add (C16:0) after the palmitic acid and C20:0 for the arachidic acid.

Line 259, in results and discussion

The same previous comment for the unsaturated fatty acids

Line 263, in results and discussion

Remove the space between the figure number and the letter

Line 264, in results and discussion

Consider adding a table with the correlation values between the principal components and the analysed parameters? This table can be added in the supplementary data files.

Line 267, in results and discussion

Use the sample abbreviations applied above, H1 – H16, between brackets, to elucidate the reader.

Figure 4, in results and discussion

In this figure consider changing the samples names in order to maintain the nomenclature applied from the beginning of the article (H1 - H16). The indication regarding the concentration of ozone can be made in the figure caption and for the fatty acids, in the matrix association, indicate the fatty acid names in the figure caption.

Line 275, in results and discussion

Change to oils’ quality

Line 276, in results and discussion

Change the fatty acids profile to saturated fatty acids total content.

Indicate between brackets which are the treated maize samples and the control samples.

Line 278, in results and discusssion

Remove “the” before lowest ozone concentration

Remove “of the application”

Line 279, in results and discussion

Add a comma after 52%

Line 282, in results and discussion

Consider removing “during determining” and start the sentence by the polyunsaturated fatty acids content in the analyzed maize hybrids....decreased more than....

Line 285, in results and discussion

Reformulate by adding “an increase in the linoleic acid content (36.7% in relation....) in the analyzed maize hybrids (H1, H11 and H14).

Lines 287 and 288, in results and discussion

Indicate between brackets which samples are the analyzed maize hybrids and which samples are the control samples

In line 288, add references and indicate between brackets which value is considered a high PUFA content associated with beneficial health effects.

Lines 289 and 290, in results and discussion

Add references to support the finding

Line 292, in results and discussion

Reformulate to "Only the total saturated fatty acids, such as palmitic and stearic acids, increased during the ozonation process." Remove “in the case of”

Line 294, in results and discussion

Remove “in the case of” and reformulate accordingly, especially linoleic acid, when the lowest ozone concentration (40 mg/L) was applied.

Line 296, in results and discussion

Remove “which is”

Line 297, in results and discussion

Change “because of degradation of PUFAs” to because of PUFAs’ degradation

Line 298, in results and discussion

Remove “of total content”

Line 299, in results and discussion

Add the “of” proposition in “decreased of unsaturated”

Line 300, in results and discussion

Change to PUFA's ozonolysis (e.g., linoleic and alfa-linolenic acids)

Line 301, in results and discussion

Change to numerous factors mentioned by Choe [59] such as…

Line 302, in results and discussion

Remove etc and give detailed information

Line 303, in results and discussion

Substitute “the oxidation of oil” to the oil oxidation

Figure 5, in results and discussion

Change the figure number, line 337, from Figure 3 to Figure 5

Line 341, in results and discussion

Change “such as fatty acids occurred during ozone treatment” to such as fatty acids content after ozone treatment

Line 342, in conclusions

Put “except for H2” between commas

Reformulate accordingly: Results of ozone application indicated that, for all the determined mycotoxins in the maize samples, the ozone treatment (indicate which concentration) successfully reduced mycotoxins' content to values below the LOQ (indicate the LOQ value).

Line 344, in conclusions

If the values are below the LOQ, the authors shouldn´t say that the “hybrids are mycotoxin-free”. It is clearer if you add the range of the determined values.

Line 345, in conclusions

“an increase of MUFA and a decrease of PUFA”

Line 346, in conclusions

“this changes could be a hybrid specific”. Can you make clearer.

Line 347, in conclusions

Give examples of the parameters that should be optimized

References section

Confirm the references format (e.g., 16, 36, 43, 44, 45, 49, 51 and 60)

Author Response

Replies for

Referee 1

Authors comments

Thank you very much for your consideration and invaluable comments on our recent revision. Your suggestions and feedback on the content helped us to improve the manuscript. We appreciate your effort and the time you took to read the manuscript. We carefully went through all of your comments and revised the manuscript accordingly. We have revised, taking all your comments into account, added content and re-organized the rewriting.

Text Location

Comment

Lines 50 and 51 in Introduction section

We agreed with referee’s opinion to include  the range of temperatures after temperate regions, warm regions and high soil temperatures. It is rewritten in lines 50-51“ …..Europe and China [8], while aflatoxins 50 (AFs) are prevalent in tropical and sub-tropical regions because fungus needs temperatures of 32-38 °C for infection [9].“

Line 55 and line 57, in introduction section

According to another referee’s observation second paragraph of Introduction is divided into two section. First about mycotoxins occurrence, and second about mycotoxins toxicity.  Toxicity for the different mycotoxins are in lines 59, 60, 61, 62, 64 and 65: “Mycotoxins have different effects on human and animal health. DON greatly affects the intestine and the immune system of pigs [15], decreasing feed consumption at doses from 0.6 to 3 mg DON/kg feed [16]. “, “Although ZEN has lower acute toxicity in comparison with many other mycotoxins (oral LD 50 values of >2000–20 000 mg/kg b.w. in mice, rats and guinea pigs),…” and “with reported LD50 for the adult, male Fisher rat of 1.2 mg aflatoxin B1/kg body weight [20].”

Line 70, in introduction section

We agreed with referee’s observation that examples should be added. It is added in Lines 78-82. “Wang et al. [38] managed to reduce DON content in wheat using 78 ozone at level of 75 ppm, obtaining 53.48% reduction rate after 90 min…….”

Line 73, in introduction section

We agreed with referee’s opinion that quality parameters affected by ozone should be added. Therefore line 73 is completely rewritten into line 84 “Furthermore, some studies have found that ozonization can cause changes in grain quality 84 attributes, such as protein [38], fatty acids [35, 40], rheological properties [41], color [40], phenolic compounds and antioxidant ability [35], especially in cereals and its products.”

Line 75, in introduction section

We agreed with the referee's opinion that a whole paragraph should have been written on the impact of ozone on quality parameters. Unfortunately, we did not find enough researchs for the entire paragraph. That's why we wrote line 84 about it.

Line 83, in materials and methods

The “common agriculture practice” is explained is into paragarph’s Field trial that is completely rewritten beginning in line 95  “The experiment was carried out on a chernozem soil, with the humus content of 2.86%, pH= 7.04….”

Line 89, in materials and methods

This is the reasons - The scale was reduced in order to avoid the problem with structural zeros that occur when we do not have plants in the categories of the scale, and this creates a problem when evaluating the parameters in the statistical model. For the needs of work and testing of hybrids in breeding, this "reduced" grading scale is quite sufficient.

Line 108, in materials and methods

We are decided that stay flowrate because of in many studies about ozonization  we found this term.

Line 109, in materials and methods

According referee’s observation the references and justification for applied ozone concentration. Line 127….” Similar ozone levels were applied by Wang et al. [37] and Qi et al. [40].”

Line 121, in materials and methods

According referee’s opinion we changed “evaporated just to dryness” to “evaporated until dryness.”

Line 125, in materials and methods

We agreed with the referee's opinion about the LOD and LOQ for the different evaluated Afs. It begun in line 144 “Detection limits (LODs) were 0.3 μg/kg for Afs, 1.1 μg/kg for ZEN and 22.2 μg/kg…….”.

Line 140, in materials and methods

According referee’s observation we changed “by the vortex” to “in the vortex.” Line 163

Line 141, in materials and methods

We agreed with the referee's opinion  and rewrited “after for an hour standing” to after “one hour standing, the aliquot….” Line 164

Line 147, in materials and methods

According referee’s observation it is explained identification by adding the Kovat index of the compounds Lines 168 and 169. “Identification and quantification of individual FAME……….”

The line 172 is Rewriten into” The results of the individual fatty acids content were expressed as g/100g of oil”. Yes you understood it correctly.

Line 177, in materials and methods

According referee’s observation full stop is added.  Line 199 “….correlation analysis [54]. For example….”

Line 180, in materials and methods

We agreed with the referee's opinion and the line 202 is changed into :“The principal component analysis (PCA) is a widely used multivariate technique for data reduction analysis.”

Line 185, in materials and methods

According referee’s observation reference [54] is added. Line 207

Lines 191 and 192, in results and discussion

We agreed with the referee's opinion and the lines 213-215 the temperatures conditions and precipitation levels in July and August 2021 were  added “Very warm and rainy July (heat wave from 6 ……”

Line 192, in results and discussion

According referee’s suggestion in line 215 comma before “all genotypes are affected…” is added.

Line 194, in results and discussion

The line 217 is reformulated “Our study contained three standard….”. because the English language in the manuscript  has been corrected by native speaker.

Line 195, in results and discussion

As a above line 217 is reformulated.

Line 196, in results and discussion

We agreed with the referee's opinion and substituted “rest of studied hybrids” by “remaining studied hybrids (Figure1).” Line 218

Caption of Figure 1

We agreed with the referee's opinion and substituted “rest of studied hybrids” by remaining studied hybrids. Line 222

Line 207, in results and discussion

According referee’s observation comma is added. Line 229 “Except for H2, all studied hybrids…” is added.

Line 213, in results and discussion

Weather conditions were such that only strains of fungi that produce only AFs and DON and not ZEN were developed.

Table 1, in results and discussion

The different abbreviation of mycotoxins are unified in hole manuscript. 

The LOQ values are given below the table, Line 238.

The  units is added in Table 1.

The decimals throughout the table is unified.

The caption of the table Line 237 “Mycotoxin content (average ± standard deviation) in maize hybrids before ozonation (n=4)”.

Line 220, in results and discussion

The primary goal of our manuscript is the reduction of mycotoxins in maize, not the toxicity levels that are already regulated by EU regulations.

Line 225, in results and discussion

The line 251 is reformulated into “…….crease below the LOD in all the tested samples.”

Line 226, in results and discussion

According to  the referee’s  observation line 251 is reformulated into “Ozonation at three ozone levels (40, 70 and 85 mg/L) applied for 180 minutes resulted in mycotoxins decrease below the LOD in all the tested samples”, lines 250-251.

In line 251 is added “the” in the sentence: “To the best…”

Line 235, in results and discussion

According referee’s opinion in whole paragraph percent values are shown in only one decimal and comma is in line 260 added  after the percent 70.7%. “…DON content in maize was reduced up to 70.7%, by…”

Line 238, in results and discussion

According to the referee’s  suggestion we are confirming that stay ground (line 263) and put comma “,in the study by …[26]”

Line 239, in results and discussion

According referee’s opinion in the results and discussion number of decimal is unified.

Line 244, in results and discussion

According to the referee’s suggestion  reference [60] is added line 269

Line 247, in results and discussion

According referee’s opinion number of decimals in the percentages is corrected (line 272).

Line 249, in results and discussion

According to the referee’s suggestion is accepted (line 274).

Line 252, in results and discussion

According to referee’s opinion LOQ values are added (line 277).

Line 258 and 259, in results and discussion

Abbreviations for all listed fatty acids were introduced in the material and methods (line 170), and were later used in the text.

Line 259, in results and discussion

Abbreviations for all listed fatty acids were introduced in the material and methods (line 170), and were later used in the text.

Line 263, in results and discussion

According to referee’s opinion space is removed (line 288).

Line 264, in results and discussion

Any correlations in the context of proportional data are problematic. Therefore we did not do it.

Line 267, in results and discussion

We agreed with the referee's opinion sample abbreviations are put“…on both the left (H2, H11 and H12) and right sides (H1, H5 and H14)…” (line 292).

Figure 4, in results and discussion

By mistake, the working version of the image was put (wrong), we added the correct version (Figure 4).

Line 275, in results and discussion

We agreed with the referee's opinion and changed to “oil quality” by English language in the manuscript has been corrected by native speaker (line 300).

Line 276, in results and discussion

We agreed with the referee's opinion the sentence is reformulated into “(ozone 40, 70 and 85) and the control samples (ozone 0)” (lines 301-302).

Line 278, in results and discusssion

We agreed with the referee's suggestion and removed (line 303)

Line 279, in results and discussion

We agreed with the referee's opinion comma is added (line 304).

Line 282, in results and discussion

According to the referee’s suggestion the sentence is reformulated into “The PUFA content (C18:2) in the analyzed maize hybrids (H1, H11 and H14) treated with the lowest applied…” (line 307). 

Line 285, in results and discussion

According to the referee’s suggestion the sentence is reformulated  into “an increase in the C18:2 content (36.7% in relation to the lowest applied…” (line 309).

Lines 287 and 288, in results and discussion

We agreed with the referee's opinion and changed the sentence to “The amount of linolenic acid did not change significantly during the ozonation process in the ozonized maize hybrids (ozone 40, 70 and 85), compared to the control samples (ozone 0)” (lines 311-312).

“ …a high content of PUFA (2-3 g/day) has positive effects on human health, therefore…” (lines 313-314).

Lines 289 and 290, in results and discussion

We agreed with the referee's opinion and put reference [66] “…nutritive value of maize samples [66]” (line 314).

Line 292, in results and discussion

According to the referee’s suggestion the sentence is reformulated  “Only the total SFA, such as C16:0 and C18:0, increased during the ozonation process.” (line 316).

Line 294, in results and discussion

According to the referee’s suggestion the sentence is reformulated  to “Accordingly, especially C18:2, when the lowest concentration of ozone (40 mg/L) was applied.” (lines 316-317).

Line 296, in results and discussion

We agreed with the referee's opinion and “which is” is removed. (line 319)

Line 297, in results and discussion

According to the referee’s suggestion the sentence is reformulated  to “…because of PUFAs’ degradation…”(line 320)

Line 298, in results and discussion

We agreed with the referee's opinion and “of total content” is removed (line 321).

Line 299, in results and discussion

We agreed with the referee's opinion and “of” proposition in “decreased of unsaturated” is added. (line 322).

Line 300, in results and discussion

According to the referee’s suggestion the sentence is reformulated  to “…or PUFA’ ozonolysis (e.g. C18:2 and C18:3)..”(line 323).

Line 301, in results and discussion

We agreed with the referee's opinion and we add other factors “Numerous factors mentioned by Choe and Min [68], such as concentration and type of oxygen, and free fatty acids, mono- and diacylglycerols, pigments, antioxidants, oil processing, energy of heat or light, transition metals, peroxides and thermally oxidized compounds jointly affect oil oxidation. (lines 323-326)

Line 302, in results and discussion

We agreed with the referee's opinion and we add other factors “Numerous factors mentioned by Choe and Min [68], such as concentration and type of oxygen, and free fatty acids, mono- and diacylglycerols, pigments, antioxidants, oil processing, energy of heat or light, transition metals, peroxides and thermally oxidized compounds jointly affect oil oxidation. (lines 323-326)

Line 303, in results and discussion

We agreed with the referee's opinion “the oxidation of oil” substituted to the oil oxidation. (line 326)

Figure 5, in results and discussion

We agreed with the referee's suggestion and changed figure number (line 360)

Line 341, in results and discussion

We agreed with the referee's opinion “such as fatty acids occurred during ozone treatment” changed to such as fatty acids content after ozone treatment. (line 364)

Line 342, in conclusions

We agreed with the referee's opinion and “except for H2” is put between commas (line 365).

According to the referee’s suggestion the sentence is reformulated  “Results of ozone application indicated that,  for all the determined mycotoxins in the maize samples, the ozone treatment (indicate which concentration) successfully reduced mycotoxin content to values below the LOQ (DON (74.0 μg/kg), ZEN (3.6 μg/kg), AFs (1.0 μg/kg)).”(lines 365-367)

Line 344, in conclusions

As a above lines 365-367 is reformulated.

Line 345, in conclusions

According to the referee’s suggestion and  “an increase of MUFA and a decrease of PUFA” (line 369)

Line 346, in conclusions

We agreed with the referee's opinion and made the sentence more clearer. “However, these differences could be dependent of genotype, so the research in this area should continue.” Line (370)

Line 347, in conclusions

We agreed with the referee's suggestion and give examples into line 371.

References section

We agreed with the referee's suggestion and the references format (books, site etc.)  unified with propositions of journal

Resubmission Date

5 September 2022

Reviewer 2 Report

The present work evaluates the effect of ozone application on ground maize and the potential impact on mycotoxins concentrations. Furthermore, the fatty oils profile was evaluated (MUFA and PUFA). Three ozone concentrations were used: 40, 70, and 85 mg/L. Sixteen maize genotypes were evaluated under field conditions.

The manuscript is well written, well structured, and denotes a high-quality work in general terms. The assay design is appropriate, and the statistical section is very detailed and accurate. The main strength of the current study is the results section, with a solid discussion section. Figures, tables, and bibliography sections are appropriate too.

An interesting point of the work is that it showed novel information regarding decreasing mycotoxins contamination after the ozone application, highlighted in low ozone doses (40 mg/L). However, this decrease in mycotoxins contamination is accompanied by negative MUFA/PUFA ratio changes, which could imply lower nutritional quality. Although there are previous reports of the application of ozone, I consider that the results obtained are novel for the study area and make a significant contribution to the subject.

In order to improve the quality of the manuscript, some minor points were marked below:

-Line 35: please, replace “planted” with “sown”. Maize is a cereal crop, not a forest species.

-Line 82: please, clarify the following abbreviation: “RCB” (randomized complete block). May be is not common for all the readers, even sometimes in several works appears as “RCBD”.

-Lines 83-84: What is “a common agricultural practice”? For which agricultural region? This statement is not valid for all the worldwide cereal regions. It would be interesting if the authors could describe (briefly) more details of the field assay to ensure the assay repeatability. For instance, more details regarding sowing date, climatic conditions, soil classification, soil tillage (zero or conventional?), duration of the crop cycle, alternative hosts control, previous crop/crop rotation, etc…are needed because these agronomic factors can modify plant-pathogen interaction (changes in flowering date and inoculum exposure, and the subsequent mycotoxin production).

Furthermore, if exists information regarding the level of resistance/tolerance against Fusarium/Aspergillus for each hybrid, the authors should aggregate it. Although the phytosanitary behavior of the hybrids can be inferred from the mycotoxin values (indirectly), it would be enriching for the work to add this type of information.

-Other comments: some Oxford commas are missed throughout the entire manuscript. Furthermore, some was/were are wrong. Please add it or check it.

Author Response

Replies for

Referee 2

Authors comments

Thank you very much for your consideration and invaluable comments on our recent revision. Your suggestions and feedback on the content helped us to improve the manuscript. We appreciate your effort and the time you took to read the manuscript. We carefully went through all of your comments and revised the manuscript accordingly. We have revised, taking all your comments into account, added content and re-organized the rewriting.

-Line 35: We agreed with referee’s observation word “planted” is replaced with “sown” - line 35.

-Line 82: We agreed with referee’s opinion and replace abbreviation: “RCB” (randomized complete block). With “RCBD” - Line100.

-Lines 83-84: We agreed with referee’s observation  that term “a common agricultural practice” should be explained. Therefore, whole paragraph 2.1 Field trial is rewritten into:

“The field trial was set up in the experimental fields of the Institute of Field and Vegetable Crops, Novi Sad, Serbia in 2021 (45 o 20’14’’N, 19 o 51’44’’E, 78 m above sea level). The climate of the region is moderately continental and autumn is drier than spring. The warmest is July and rainiest June. Annual precipitation varies between 570 and 650mm. The experiment was carried out on a chernozem soil, with the humus content of 2.86%, pH= 7.04, total nitrogen 0,24%, P2O5 27,53 mg/100g and K2O 29,23mg/100g. The preceding crop was soybean in a three year rotation (soybean-maize- wheat), and conventional soil cultivation practices were applied. Sowing was done on April 14, pre-emergence herbicide was applied 5 days later and post-emergence correction at 6-7 leaf growth stage. A total of 16 maize hybrids of different genetic back-ground, designated as H1-H16, were used as entries in a Randomize Complete Block Design (RCBD) with 3 repetitions. Four row plots (10 m long and 0.75 m wide) were planted at an approximate density of 63.500 100 plants/ha.”. Lines (92-101).

Precise information about the level of resistance/tolerance against Fusarium/Aspergillus of the tested hybrids do not exist. However, some preliminary results about the check hybrids (grown on a large scale) show that all three possess pretty high level of tolerance. 

-Other comments: We agreed with referee’s suggestion. The English language in the manuscript after rewriting has been corrected by native speaker. 

Resubmission Date

5 September 2022

Reviewer 3 Report

Changes in Fusarium and Aspergillus mycotoxins content and fatty acid profile after the application of ozone in different maize hybrids

The authors collected maize samples from experimental fields to study the effects of ozone treatment on mycotoxins and fatty acid content in maize. This paper makes reasonable use of data analysis and discusses it fully, which has certain significance for agricultural production applications. However, the background of this article is insufficient and lacks the latest research progress. The experimental design is relatively thin and the experimental scheme is not detailed enough. There is a lack of other pollution parameters except mycotoxins (such as fungal species, fungal number, etc.) and detailed questionnaires and data are lacking in the assessment of the degree of decay. The changes in corn nutrients are only fatty acids, and thus the persuasiveness is weak. It is proposed to be revised and then discussed. The final decision shall be decided by the editors.

1.Abstract

(1) Key experimental results, such as the magnitude of MUFA increase and PUFA decrease, need to be supplemented.

2. Introduction

(1) It is recommended to supplement the latest data that can reflect the extent of local corn contamination.

(2) The references are too old, it is recommended to follow up the latest research progress, supplement the latest toxicity research results of mycotoxins, and try to supplement the toxicity mechanism. Supplement the latest detoxification research. The ozone degradation method is a traditional, relatively mature chemical detoxification method that has been used on a large scale. It is recommended to supplement the latest experimental data on ozone detoxification efficiency to explain the advantages of the ozone detoxification method. It is recommended to supplement the discussion with the latest detoxification methods.

(3) It is recommended to supplement the detailed mycotoxins degradation products after ozone treatment and analyze the possible action sites of ozone.

(4) There is a syntax error on line 66.

(5) Note the use of the "()" level in line 71.

2. Material and Methods

(1) The evaluation method of natural rot degree of ears needs to be supplemented in detail, including the number of people and evaluation standard. It is suggested that the evaluation form and results should be supplemented in the form of attachments.

(2) Briefly supplement the method steps or references for water content determination, quartile screening, etc.

(3) In section 2.8, line 139, the modifications made to the original method need a brief introduction.

(4) Only fatty acids were detected as a nutritional indicator of maize, which is too single. Whether to consider detecting multiple nutrients in the future?

(5) Note whether Afs in line122 should be AFs

(6) Line137, the abbreviation does not correspond to the full name. It is recommended to add the full name corresponding to the abbreviation in the correct position

(7) Check whether there is a spelling error in reacPaynetion on line141

3.  Results and Discussion

(1) Need a simple explanation why H4/H8/H16 is used as standard.

(2) The ZEN content in most samples is below the detection limit, should we consider changing the detection method with a lower detection limit? Otherwise, the results of this experiment have certain limitations and cannot be popularized in more mycotoxins.

(3) Make a brief analysis of the large differences of mycotoxin species and content in different samples (planting conditions, temperature/moisture).

(4) Figure 3. It is recommended to add separate chromatograms (combined images can briefly illustrate the results, but the colors are close and difficult to distinguish).

3. Discussion  

(1) The discussion part should avoid the brief repetition of the experimental results, and should have corresponding analysis and prospect

5. Others

(1) Pay attention to the font size of the full text needs to be revised according to the requirements of the journal.

Author Response

Replies for

Referee 3

Authors comments

Thank you very much for your consideration and invaluable comments on our recent revision. Your suggestions and feedback on the content helped us to improve the manuscript. We appreciate your effort and the time you took to read the manuscript. We carefully went through all of your comments and revised the manuscript accordingly. We have revised, taking all your comments into account, added content and re-organized the rewriting. 

1.Abstract

(1) According referee’s opinion abstract is supplemented with sentence: “Ozone treatments  increase the content of monounsaturated fatty acids (MUFA) and decrease of polyunsaturated fatty. (lines 24-25).

  1. Introduction

(1) We agreed with the referee's opinion and is written several sentences: “Since the beginning of the 21st century, problems with high mycotoxin content in corn were not registered in Serbia until 2012, which was characterized by extreme drought, and 2014, which was characterized by floods [10]. Besides weather conditions, plant density and genotype are also quite important factors contributing to the occurrence of mycotoxins in maize [11, 12] “ Lines 52-54

(2) We are looking for the references contamination of maize with mycotoxins and eventually effect of ozonisation on maize/cereals quality. The latest reference of the topic are in the manuscript ①According referee’s opinion it is added in paragraph lines 59-71. Whereas, the  toxicity mechanism were not topic of our manuscript we did not it supplemented in the manuscript. ②We were looking for the references for the ozonisation research since it was the topic of the our research.  If we were to expand the manuscript with other detoxification research we would move away from the given manuscript topic. ③If we were to expand the manuscript with other detoxification research we would move away from the given manuscript topic

(3) As we mentioned in the conclusion “…this is the first study conducted in Serbia that examined the changes on the quality parameters of maize such as fatty acid content after ozone treatment…”. Line 363. Therefore we will plan to do it in further researches.

(4) According referee’s opinion a syntax error is corrected - line 72.

(5) According referee’s opinion sentence is rewritten - line 83.

  1. Material and Methods

(1) The evaluation method of natural rot degree of ears that pathologist do in the field is done on this way. This is completely different than sensory evaluation in the sensoric laboratory . Therefore we will plan to do it in further researches when the field experiment will include experiments with artificial inoculation of fungi on maize.

(2) According referee’s opinion it is added Line 113.

(3) In section 2.8,is detailed explained the modifications made to the original method. Lines 162-168.

(4) The future research will be include to monitor other quality parameters such as: Total content of bioactive components, tocopherols, carotenoids and phenols; quality parameters: acid number (Kbr), peroxide number (Pbr) and anisidine number (Abr), oxidative value (OV), content of conjugated dienes (KD) and conjugated trienes (KT) and sensory characteristics of the analyzed samples.

(5) According to all Reviewer’s instructions observation the abbreviation of mycotoxins, is unified during the whole manuscript.

(6) According  to referee’s opinion  the abbreviation is corrected into “…..ionization detector (GC-FID) after derivatization to their volatile FA methyl esters (FAME).”line 160

(7) According referee’s opinion the spelling error is corrected “The reaction is complete upon dissolution of the oil.” line164

  1. Results and Discussion

(1) H4/H8/H16 were used as standards because all three are hybrids grown on a large scale. At the same time, according to preliminary observation, all three possess pretty high level of tolerance.

(2) It is not necessary to do it since now in manuscript is included the LOQ for ZEN (3.6 μg/kg).

(3) Long-term monitoring of mycotoxins on maize will be the subject of future researches.

(4) We are of the opinion that the current presentation is more credible in Figure 3. Because it is easy to be seen results before and after ozonisation.

  1. Discussion

(1) We are of the opinion that based on the obtained results it was not possible to have a more detailed discussion. A detailed discussion will be done after the completion of the planned extended experiments. 

  1. Others

(1) We are corrected everything we noticed 

Resubmission Date

5 September 2022

Reviewer 4 Report

I have reviewed the manuscript entitled “Changes in Fusarium and Aspergillus mycotoxins content and fatty acid profile after the application of ozone in different maize hybrids” and found it is easy to follow.

This is an interesting report, which provides new information and issues for further exploring the management of Fusarium mycotoxin contamination. Mycotoxins are commonly detected contaminates in cereal grains, nuts, and fruits, etc. However, the sections of Introduction and Results and discussion need to be revised carefully before considering for publication.

All the marked places in the PDF version must be revised carefully.

Specific comments:

1. Overall, there are many grammatical mistakes in the whole text. I strongly suggest that the authors ask for an English language and style editing from native English speaker.

2. To a certain degree, I don’t agree with the use of “profile” in the text. In my opinion, the fatty acid profile changes should be concluded according to the analyses of all kinds of fatty acids contained in maize. However, only several major fatty acids were mentioned in the text.

3. What kind of material was used for decontamination analysis, row maize grains or ground maize? I suggest the authors to clarify this in the title and abstract. Moreover, the grammatical mistakes should be revised in the title.

4. The introduction should be rewritten especially the logic of the second paragraph needs to be rearranged, Lines 43-65, Page1-2. I suggest separating the second paragraph into two paragraphs.

5. Lines 35-36, revised the sentences to “An average of total maize production in the last decade amounted to 6,286 thousand tons in Serbia [2]”.

6. Correct the grammatical mistakes in Lines 40-42 marked with yellow.

7. Line 43, please explain FAO, and revised the sentences to “more than 25% of cereals are contaminated with mycotoxins worldwide”.

8. The abbreviation of mycotoxins, such as ZEN and ZEA, AFs and Afs, should be uniform throughout the whole text. Are ZEN and ZEA the abbreviations of different mycotoxins?

9. Line 54, please explain the abbreviations AFG1 and AFG2.

10. Lines 52-53, to the best of my knowledge, I think the description of “while ZEA is produced by F. roseum, F. tricinctum, F. sporotrichioides, F. oxysporum and F. moniliforme [10]” is not accurate or even wrong though it comes from reference. If the authors prefer to use this statement, please provide the original references here or revise it.

11. Lines 61, (10, 17–23) should be [10, 17–23].

12. Line 82, please explain RCB.

13. Line 215, Table 1, please explain B1+G1+B2+G2.

14. Line 301, Choe [59] should be Choe and Min [59].

15. The resolution of all the Figures should be improved. 

Author Response

Replies for

Referee 4

Authors comments

Thank you very much for your consideration and invaluable comments on our recent revision. Your suggestions and feedback on the content helped us to improve the manuscript. We appreciate your effort and the time you took to read the manuscript. We carefully went through all of your comments and revised the manuscript accordingly. We have revised, taking all your comments into account, added content and re-organized the rewriting.

Specific comments:

  1. We agreed with referee’s suggestion. The English language in the manuscript after rewriting has been corrected by native speaker.
  2. We agreed with referee’s opinion and instead of “profile” we used words “composition” (in Title) “dominated Fatty acids” or “fatty acids content” (result and discussion) in whole text.
  3. We specified in material and methods that its ground maize. The reason why we did not do it in the Title (Changes in Fusarium and Aspergillus mycotoxins content and fatty acids composition after the application of ozone in different maize hybrids) and abstract is as follows: This is not a common practice when examining cereal processing. Example when it is examine wheat quality of different cultivars in title and abstract of manuscripts (previous studies that have been published so far) is not specified is it meal, flour or kernel examine.
  4. We agreed with referee’s suggestion, the introduction is rewritten and the second paragraph is divided into two paragraph (Lines 43-58, Lines 59-71).
  5. We agreed with the referee's opinion and rewritten according to Reviewer’s instructions (Lines 35-36).
  6. We agreed with the referee's statement and rewritten to “…..which increase in concentration during pre-harvest, postharvest and storage [4].” Lines 40-42.
  7. Line 43, According to Reviewer’s instructions the sentence is changed into “According to the Food and Agriculture Organization (FAO) estimates, more than 43 25% of the world’s cereal production is contaminated with……”
  8. According to Reviewer’s instructions observation the abbreviation of mycotoxins, is unified during the whole manuscript.
  9. As we mentioned above the abbreviation of mycotoxins, is unified during the whole manuscript.

10.We agreed with the referee's opinion and whole paragraph is edited with supplemented data this. (Lines 59-71).

  1. According to Reviewer’s instructions “brackets are changed. Lines 66-67.
  2. Lines 83-84: We agreed with referee’s observation about RCB. Therefore, whole paragraph 2.1 Field trial is rewritten into:

“The field trial was set up in the experimental fields of the Institute of Field and Vegetable Crops, Novi Sad, Serbia in 2021 (45 o 20’14’’N, 19 o 51’44’’E, 78 m above sea level). The climate of the region is moderately continental and autumn is drier than spring. The warmest is July and rainiest June. Annual precipitation varies between 570 and 650mm. The experiment was carried out on a chernozem soil, with the humus content of 2.86%, pH= 7.04, total nitrogen 0,24%, P2O5 27,53 mg/100g and K2O 29,23mg/100g. The preceding crop was soybean in a three year rotation (soybean-maize- wheat), and conventional soil cultivation practices were applied. Sowing was done on April 14, pre-emergence herbicide was applied 5 days later and post-emergence correction at 6-7 leaf growth stage. A total of 16 maize hybrids of different genetic back-ground, designated as H1-H16, were used as entries in a Randomize Complete Block Design (RCBD) with 3 repetitions. Four row plots (10 m long and 0.75 m wide) were planted at an approximate density of 63.500 100 plants/ha.”. Lines (92-101).

  1. According to Reviewer’s instructions it is change into “AFB1+AFG1+AFB2+AFG2” Table 1. Line 237
  2. We agreed with the referee's statement and rewritten to “Choe and Min [68]”.
  3. The Figures are of poorer quality due to the file size. Figures were sent to the publisher in JPEG format with a resolution according to the magazine's requirements

Resubmission Date

5 September 2022

Round 2

Reviewer 3 Report

1.Abstract

(1) Key experimental results, such as the magnitude of MUFA increase and PUFA decrease, need to be supplemented.

2. Introduction

(1) It is recommended to supplement the latest data that can reflect the extent of local corn contamination.

(2) The references are too old, it is recommended to follow up the latest research progress,  supplement the latest toxicity research results of mycotoxins, and try to supplement the toxicity mechanism.  Supplement the latest detoxification research. The ozone degradation method is a traditional, relatively mature chemical detoxification method that has been used on a large scale. It is recommended to supplement the latest experimental data on ozone detoxification efficiency to explain the advantages of the ozone detoxification method.  It is recommended to supplement the discussion with the latest detoxification methods.

(3) It is recommended to supplement the detailed mycotoxins degradation products after ozone treatment and analyze the possible action sites of ozone.

(4) There is a syntax error on line 66.

(5) Note the use of the "()" level in line 71.

3. Material and Methods

(1) The evaluation method of natural rot degree of ears needs to be supplemented in detail, including the number of people and evaluation standard. It is suggested that the evaluation form and results should be supplemented in the form of attachments.

(2) Briefly supplement the method steps or references for water content determination, quartile screening, etc.

(3) In section 2.8, line 139, the modifications made to the original method need a brief introduction.

(4) Only fatty acids were detected as a nutritional indicator of maize, which is too single. Whether to consider detecting multiple nutrients in the future?

(5) Note whether Afs in line122 should be AFs

(6) Line137, the abbreviation does not correspond to the full name. It is recommended to add the full name corresponding to the abbreviation in the correct position

(7) Check whether there is a spelling error in reacPaynetion on line141

4.  Results and Discussion

(1) Need a simple explanation why H4/H8/H16 is used as standard.

(2) The ZEN content in most samples is below the detection limit, should we consider changing the detection method with a lower detection limit? Otherwise, the results of this experiment have certain limitations and cannot be popularized in more mycotoxins.

(3) Make a brief analysis of the large differences of mycotoxin species and content in different samples (planting conditions, temperature/moisture).

(4) Figure 3. It is recommended to add separate chromatograms (combined images can briefly illustrate the results, but the colors are close and difficult to distinguish).

5. Discussion  

(1) The discussion part should avoid the brief repetition of the experimental results, and should have corresponding analysis and prospect

6. Others

(1) Pay attention to the font size of the full text needs to be revised according to the requirements of the journal.

Author Response

Replies for

Referee 3

Authors comment

Thank you very much for your consideration and invaluable comments on our recent revision. As we now received the same comments in Round 2 as in Round 1, for which we sent detailed answers on September 5. Therefore, we are sending the same answers for Round 2 in the attachment.

1. Abstract

(1) According referee’s opinion abstract is supplemented with sentence: “Ozone treatments  increase the content of monounsaturated fatty acids (MUFA) and decrease of polyunsaturated fatty. (lines 24-25).

  1. Introduction

(1) We agreed with the referee's opinion and is written several sentences: “Since the beginning of the 21st century, problems with high mycotoxin content in corn were not registered in Serbia until 2012, which was characterized by extreme drought, and 2014, which was characterized by floods [10]. Besides weather conditions, plant density and genotype are also quite important factors contributing to the occurrence of mycotoxins in maize [11, 12] “ Lines 52-54

(2) We are looking for the references contamination of maize with mycotoxins and eventually effect of ozonisation on maize/cereals quality. The latest reference of the topic are in the manuscript ①According referee’s opinion it is added in paragraph lines 59-71. Whereas, the  toxicity mechanism were not topic of our manuscript we did not it supplemented in the manuscript. ②We were looking for the references for the ozonisation research since it was the topic of the our research.  If we were to expand the manuscript with other detoxification research we would move away from the given manuscript topic. ③If we were to expand the manuscript with other detoxification research we would move away from the given manuscript topic

(3) As we mentioned in the conclusion “…this is the first study conducted in Serbia that examined the changes on the quality parameters of maize such as fatty acid content after ozone treatment…”. Line 363. Therefore we will plan to do it in further researches.

(4) According referee’s opinion a syntax error is corrected - line 72.

(5) According referee’s opinion sentence is rewritten - line 83.

  1. Material and Methods

(1) The evaluation method of natural rot degree of ears that pathologist do in the field is done on this way. This is completely different than sensory evaluation in the sensoric laboratory . Therefore we will plan to do it in further researches when the field experiment will include experiments with artificial inoculation of fungi on maize.

(2) According referee’s opinion it is added Line 113.

(3) In section 2.8,is detailed explained the modifications made to the original method. Lines 162-168.

(4) The future research will be include to monitor other quality parameters such as: Total content of bioactive components, tocopherols, carotenoids and phenols; quality parameters: acid number (Kbr), peroxide number (Pbr) and anisidine number (Abr), oxidative value (OV), content of conjugated dienes (KD) and conjugated trienes (KT) and sensory characteristics of the analyzed samples.

(5) According to all Reviewer’s instructions observation the abbreviation of mycotoxins, is unified during the whole manuscript.

(6) According  to referee’s opinion  the abbreviation is corrected into “…..ionization detector (GC-FID) after derivatization to their volatile FA methyl esters (FAME).”line 160

(7) According referee’s opinion the spelling error is corrected “The reaction is complete upon dissolution of the oil.” line164

  1. Results and Discussion

(1) H4/H8/H16 were used as standards because all three are hybrids grown on a large scale. At the same time, according to preliminary observation, all three possess pretty high level of tolerance.

(2) It is not necessary to do it since now in manuscript is included the LOQ for ZEN (3.6 μg/kg).

(3) Long-term monitoring of mycotoxins on maize will be the subject of future researches.

(4) We are of the opinion that the current presentation is more credible in Figure 3. Because it is easy to be seen results before and after ozonisation.

  1. Discussion

(1) We are of the opinion that based on the obtained results it was not possible to have a more detailed discussion. A detailed discussion will be done after the completion of the planned extended experiments. 

  1. Others

(1) We are corrected everything we noticed 

Resubmission Date

8 September 2022

Reviewer 4 Report

Overall, the whole text has been revised according to reviewers’ comments.

However, in my opinion, the description of “while ZEA is produced by F. roseum, F. tricinctum, F. sporotrichioides, F. oxysporum and F. moniliforme [13]” is wrong! So far as I know, the two species F. oxysporum and F. moniliforme can not produce zearalenone. I strongly advise the authors to revise this sentence. The current statement is totally unacceptable to me.

Author Response

Replies for

Referee 4

Authors comment

We would like to thank Reviewer 2 for his expert comment and suggestion, and we appreciate your time and effort because your comment helped us improve our manuscript.

Specific comment:

  1. We agreed with the referee's opinion and the sentence is rewrited into “…while ZEN is produced by F. roseum, F. tricinctum and F. sporotrichioides [13].” (line 56)

Resubmission Date

8 September 2022
